# Optimizing the Local Charge of Graphene via Iron Doping to Promote the Adsorption of Formaldehyde Molecules—A Density Functional Theory Study

Xiao Zhang [1], Chen Chen [2,*], Ting Cheng [1], Yizhuo Yang [2], Jiaxin Liu [2], Jiarui Zhu [2], Baoxuan Hou [2], Xin Xin [2] and Mingyue Wen [2]

1   School of Environmental Ecology, Jiangsu City Vocational College, Nanjing 210017, China; zhangxiao@jsou.edu.cn (X.Z.); wnchengting@sina.com (T.C.)
2   School of Environmental and Chemical Engineering, Jiangsu University of Science and Technology, Zhenjiang 212100, China; hznuyyz@163.com (Y.Y.); ljxdhr012@163.com (J.L.); zjr000323@163.com (J.Z.); hbx.1999@outlook.com (B.H.); xin_xiny@163.com (X.X.); mingyuewen1998@hotmail.com (M.W.)
*   Correspondence: chenc@just.edu.cn; Tel./Fax: +86-0511-85639001

**Abstract:** Formaldehyde is a colorless, pungent, and highly volatile toxic gas known for its detrimental effects on the brain, respiratory, and nervous systems. The adsorption method emerges as an effective approach for detecting and mitigating formaldehyde gas, with the adsorption material serving as its core component. Graphene, a two-dimensional nanomaterial with remarkable properties, exhibits enhanced adsorption capabilities when subjected to metal doping, which alters its local geometric and charge characteristics. In this investigation, theoretical first-principles density functional technology was employed to optimize the efficiency of Fe-doped graphene in formaldehyde adsorption. The calculated adsorption bond length and energy were used to determine the type of adsorption. Then, the calculated Bader charge, density of states (partial density of states), and differential valence charge density distribution were used to analyze the electron transfer process before and after adsorption. Finally, the theoretical optical properties analysis result was applied to analyze the potential of Fe-doped graphene for formaldehyde detection. The findings indicated that Fe-doped graphene constitutes a viable and stable doping structure, accompanied by a notable shift in valence charge distribution around the doped iron atom. This altered charge distribution facilitated the chemical adsorption process, leading to reduced adsorption spacing and increased adsorption energy. Throughout the chemical adsorption process, there was evident charge transfer between carbon (formaldehyde) and iron atoms, as well as between oxygen (formaldehyde) and iron atoms. The formation of adsorption bonds primarily involved the p-orbital electrons of carbon and oxygen atoms, along with the p- and d-orbital electrons of iron atoms. Ultimately, the Fe-doped graphene material exhibited promising applications in the realm of formaldehyde molecular detection, marked by significant theoretical disparities in optical properties before and after the adsorption process.

**Keywords:** adsorption; graphene; density functional theory; formaldehyde; Fe doping

## 1. Introduction

Formaldehyde, a ubiquitous small-molecule organic compound, finds extensive application in various sectors such as chemical production, medicine, health, and furniture decoration. Recognized for its potential to inflict significant harm on organisms and humans, formaldehyde is associated with acute mucosal (skin) irritation, chronic physical immune system dysfunction, carcinogenicity, and teratogenicity [1,2]. Consequently, the World Health Organization has designated formaldehyde as a class 1 carcinogen and categorized it as a toxic and harmful water pollutant (class 1) [3,4]. Unlike traditional gaseous pollutants such as NO, $NO_2$, and $SO_2$, formaldehyde has become one of the main indoor air pollutants due to the use of substandard furniture and decoration materials.

Long-term exposure to such indoor air could cause irreversible harm to the body [4]. Due to the colorless nature of formaldehyde and the difficulty in perceiving its odor at low concentrations, it was difficult for the human body to perceive it, thereby causing harm to the human body [5]. The rapid and accurate detection of formaldehyde is crucial for the timely detection of indoor air quality abnormalities and the prevention of long-term exposure. Therefore, it is necessary to develop efficient formaldehyde gas detection methods to ensure indoor air safety and maintain physical health. Consequently, the detection and mitigation of formaldehyde emerge as critical research areas in the realm of environmental protection. Currently, mature formaldehyde detection methods include gas chromatography [6], polarography [7] and the fluorometry method [8]. Due to the large equipment size, complex detection methods, and high cost, all these methods were not suitable for long-term, large-scale, and simple indoor formaldehyde detection. Therefore, the development of formaldehyde detection technology with simple equipment, convenient operation, and low cost is worth studying [9,10]. So far, the field acknowledges the significant potential of adsorption technology, characterized by a straightforward principle and cost-effective equipment [11–13]. At its core lies the imperative development of efficient adsorption materials, positioning adsorption as a promising technology for formaldehyde treatment and detection. A series of studies have shown that carbon materials, including graphene [14] and carbon nanotubes [15], and their improved materials [16,17] have acceptable adsorption performance for formaldehyde.

As mentioned earlier, graphene, as the earliest extensively studied two-dimensional nanocarbon material, holds substantial promise in the adsorption [18,19] and detection of small molecule gases [20–22], owing to its expansive specific surface area, heightened conductivity, and rapid photoelectric response [23,24]. Current research has unequivocally demonstrated graphene's potential as an efficient adsorbent for harmful gases, attributable to its surface interaction with numerous small molecule gases. To further augment the surface adsorption process, the incorporation of impurity elements into graphene molecules has been recognized as an effective and viable strategy [25]. Tang, for instance, demonstrated that transition metal-doped graphene could enhance the adsorption of HCl molecules [26]. Jia et al. proposed that noble metal (Ag, Pt, and Au)-doped graphene exhibited the capacity to adsorb $NO/NO_2$ molecules [27]. Seyed-Talebi's research affirmed that Al- and B-doped graphene effectively enhanced the adsorption of $NH_3$ molecules [28]. Padilla et al. established that the adsorption process could be reinforced via the doping of P and Si atoms on graphene [29]. Additionally, Zhang's work suggested that vacancy defects in Ti-doped graphene could promote the adsorption process of $SO_2$ molecules [30].

Moreover, the adsorption of gaseous small molecules on doped graphene frequently entails substantial electron transfer, consequently yielding detectable optical responses. This characteristic renders doped graphene a viable candidate for efficient small-molecule gas detection materials. Wang et al., for instance, proposed that changes in optical parameters could be discerned during the adsorption process of methanol molecules on B- or N-doped graphene [31]. Zhang et al. indicated that significant variations in electronic and optical properties could be observed when phosgene molecules are adsorbed on transition metal-doped graphene [32]. Muhammad's research demonstrated a noteworthy red shift in the visible range upon hydrogen adsorption on nitrogen-doped graphene [33]. Additionally, Zhang et al. suggested that gas adsorption on Pd-doped graphene could induce variations in optical absorption spectra [34].

In an era of rapid development, computational chemistry has gained widespread recognition as a potent research method for exploring reaction processes, material structures, and their photo-electric effects [35,36]. Among the various methodologies, density functional theory (DFT) based on first principles stands out as one of the most effective and accurate theoretical calculation methods. Its extensive application in the theoretical research arena of nano-2D carbon materials attests to its prominence [37,38]. In line with this, the Vienna Ab-initio Simulation Package (VASP) software (5.4.4), employing the DFT method, is used for investigating the adsorption process of formaldehyde molecules on

both graphene and iron-doped graphene substrates. In this work, the procedure involved the construction of initial configurations for both graphene (GH) and iron-doped graphene (Fe-GH), followed by the determination of stable structures via calculations. Subsequently, multiple initial formaldehyde adsorption configurations were generated, and optimal adsorption configurations were selected based on the calculation of adsorption energy. Furthermore, an exploration of the electronic properties and transfer processes of the optimal adsorption configuration was undertaken. Finally, the theoretical optical properties of the optimal configurations were calculated. Notably, iron doping effectively altered the local charge distribution of the graphene structure. The adsorption process manifested as typical chemical adsorption, characterized by significant charge transfer and the formation of stable adsorption chemical bonds between doped iron atoms and formaldehyde molecules. The marked alterations in theoretical optical properties before and after adsorption position it as a promising material for the detection of formaldehyde molecules.

## 2. Computational Methods and Models

This investigation employed the density functional theory (DFT) method grounded in first principles for the systematic computation of pertinent properties. Initially, the construction of relevant models was undertaken using VESTA software (3.4.7). Subsequently, the Vienna Ab-initio Simulation Package (VASP) assumed a central role in the primary computational research. The Perdew–Burke–Ernzerhof functional (PBE) [39,40] within the generalized gradient approximation (GGA) method [41,42] was employed to compute the exchange-correlation potential. Critical computational parameters included cut-off energy set at 450 eV and K-points grids with dimensions of $4 \times 4 \times 4$ (surface model calculations utilized a $4 \times 4 \times 1$ grid) during the computation process. The convergence criterion for energy was established at $10^{-5}$ eV. This calculation accuracy had been applied in many DFT computational literature [43–45], and then the K-point grid size and cut-off energy were tested at the initial calculation. Preceding further calculations, all theoretical structures underwent thorough optimization [46,47]. The primary calculated properties encompassed the optimal adsorption model, adsorption energy, Bader charge, valence charge distribution, density of states, and optical properties. The formula for computing the formation energy of iron-doped graphene is as follows:

$$\Delta E_f = E_{Fe\text{-}GH} - E_{GH} \qquad (1)$$

Among them, $\Delta E_f$ is the formation energy of Fe-doped graphene; $E_{Fe\text{-}GH}$ is the energy of Fe-doped graphene; and $E_{GH}$ is the energy of graphene.

The formula for calculating the adsorption energy of formaldehyde molecules on the original graphene is as follows:

$$\Delta E_{ad\text{-}CH} = E_{ad\text{-}GH} - (E_{GH} + E_f) \qquad (2)$$

Among them, $\Delta_{ad\text{-}CNT}$ is the adsorption energy of formaldehyde molecules on the original GH; $E_{ad\text{-}GH}$ is the energy of optimized adsorption configuration of formaldehyde molecules on the original GH; $E_{GH}$ is the energy of CNT; and $E_f$ is the energy of free formaldehyde molecules.

The calculation formula for the adsorption energy of formaldehyde molecules on Fe doped graphene is as follows:

$$\Delta E_{ad\text{-}Fe\ doped\ GH} = E_{ad\text{-}Fe\ doped\ GH} - \left(E_{Fe\ doped\ GH} + E_f\right) \qquad (3)$$

Among them, $\Delta_{ad\text{-}Fe\ doped\ GH}$ is the adsorption energy of formaldehyde molecules on Fe-doped GH. $E_{ad\text{-}Fe\ doped\ GH}$ is the energy of optimized adsorption configuration of formaldehyde molecules on Fe-doped graphene; $E_{Fe\ doped\ GH}$ is the energy of Fe-doped GH; and $E_f$ is the energy of free formaldehyde molecules.



The original calculation model and structural parameters applied in this study are as follows. Firstly, this study commenced by establishing a periodic model of graphene (GH) (Figure 1a) and Fe-doped graphene (Fe-GH) (Figure 1b), along with a molecular model of formaldehyde (FM) (Figure 2a). The graphene model, as illustrated in Figure 1a, is a single-layer two-dimensional material, and was derived by cutting the 001 faces of conventional graphite crystal cells (Figure 2b) and expanding the cells to dimensions of $4 \times 4 \times 1$. A 10 Å vacuum layer was incorporated to mitigate vertical periodic interactions [48]. The lattice parameters for GH and Fe-GH structures were set as a = 9.87 Å, b = 9.87 Å, c = 10 Å and $\alpha$ = 90°, $\beta$ = 90°, $\gamma$ = 120°. The GH structure comprised 32 carbon atoms, and the original Fe-GH structure (Figure 1b) was obtained by substituting a carbon atom with an iron atom in the graphene model. Secondly, the formaldehyde was modeled (Figure 2) according to its known chemical structure, incorporating a bond length of approximately 1.09 Å between carbon and hydrogen atoms, a bond length of approximately 1.20 Å between carbon and oxygen atoms, a bond angle of approximately 111.5° for H-C-H atoms, and a bond angle of approximately 121.8 degrees for H-C-O atoms. Drawing upon prior research, four relatively stable structures were constructed for the formaldehyde adsorption process on GH (Figures S1–S4) and five for Fe-GH (Figures S5–S9).

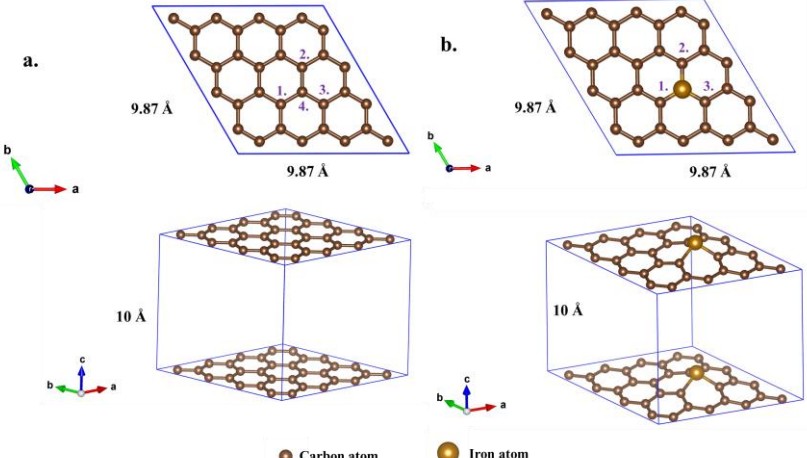

**Figure 1.** The initial monolayer graphene configuration: (**a**) GH- and Fe-doped graphene configuration. (**b**) Fe-GH; the purple number represented the number of carbon atoms. (the three arrows in the bottom left corner represented the lattice direction and same throughout the paper).

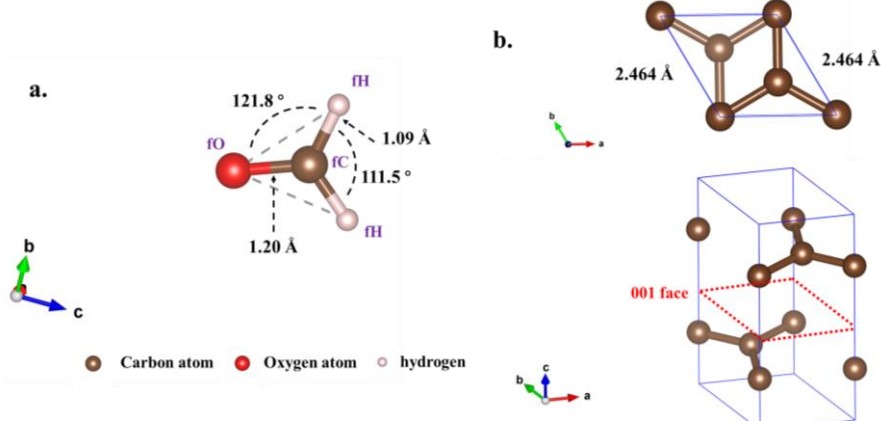

**Figure 2.** The formaldehyde molecular model (**a**). The atomic numbers in formaldehyde molecules were fO, fC, and fH for oxygen, carbon, and hydrogen atoms, respectively, and the original graphite molecular model (**b**).

## 3. Results and Discussions

### 3.1. GH and Fe-GH Structural Optimization

Prior to embarking on the formaldehyde adsorption calculations, the initial models of both GH and Fe-GH underwent optimization, with the structural optimization results presented in Figure 3. As depicted in Figure 3a, the GH model exhibited the characteristic hexagonal structures indicative of sp2 hybridization, with all the C-C bond lengths of approximately 1.424 Å. All the nearly equivalent C-C bonds facilitated unrestricted movement of the remaining valence electrons of carbon atoms throughout the entire graphene plane, thereby engendering an almost uniform distribution of valence charges across the entire graphene plane (Figure 3c).

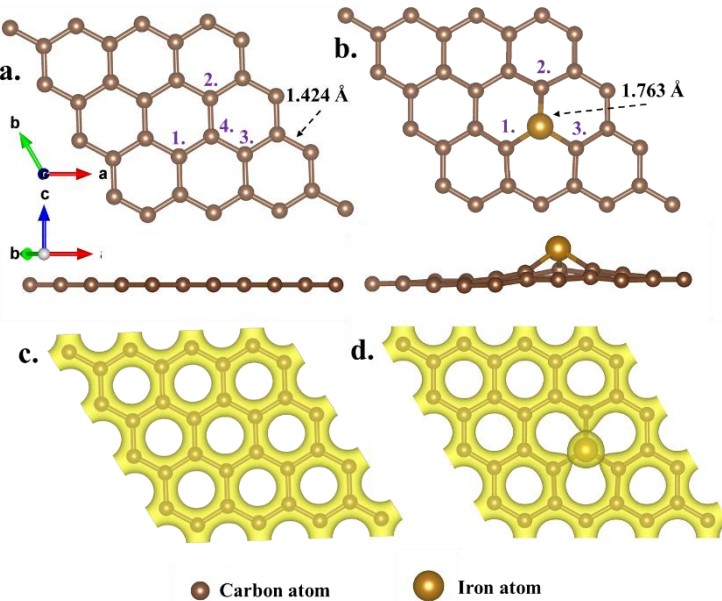

● Carbon atom ● Iron atom

**Figure 3.** The optimized GH (**a**) and Fe-GH (**b**) substrate configurations and their charge distribution (**c**,**d**); the purple number represents the number of carbon atoms.

Following the doping process, the structural integrity of the graphene hexagon persisted, and the iron atom demonstrated stable integration among carbon atoms. Subsequently, all the C1-Fe, C2-Fe, and C3-Fe bond lengths in Fe-GH measured approximately 1.763 Å (Figure 3b). The elongation of the bond lengths resulted in the iron atom positioning slightly higher than the plane of graphene molecules. In this configuration, valence charges were uniformly distributed across the graphene surface, exhibiting a spherical concentration near the iron atoms (Figure 3d). This concentrated charge distribution markedly influenced the adsorption process of formaldehyde molecules. Upon calculation, the energy values for the original GH and Fe-GH structures were approximately −295.57 and −288.97 eV, respectively, yielding a formation energy for Fe-GH of approximately 6.6 eV. Reference values indicated that when the formation energy of metal-doped graphene-like materials fell within the range of 6 to 8 eV, the doped material could be considered relatively stable [32,49]. Thus, it could be asserted that the Fe-doped graphene model was sufficiently reliable and stable.

### 3.2. Adsorption Properties of Formaldehyde on Pristine Graphene

Building upon structural optimization, the adsorption configuration of formaldehyde molecules on the GH surface was systematically examined. The optimized structures of the four fundamental configurations are presented in Figure 4 and Table 1. Notably, the adsorption distances for the four optimized configurations were measured at 3.49 Å (a. fO-C4), 3.50 Å (b. fO-C1), 3.79 Å (c. fO-C3), and 3.60 Å (d. fH-C2), respectively. Considering that the typical length of chemical bonds was generally below 3 Å, it could be deduced that

no stable chemical bond formed between formaldehyde molecules and the GH surface. The adsorption process could thus be attributed to physical adsorption. Table 1 further illustrates the adsorption energies of the four configurations, computed using Equation (2). As delineated in Table 1, the adsorption energies for these configurations were recorded as −0.0255, −0.0230, −0.0226, and −0.0282 eV. The negative values of the adsorption energies signified the proclivity of formaldehyde molecules to be adsorbed in proximity to the graphene substrate. Notably, the adsorption energies, all approximating −0.02 eV, indicated that van der Waals intermolecular forces exerted a predominant influence in the adsorption process [50]. Furthermore, Figure 5 illustrates the valence charge distribution of formaldehyde adsorbed on the GH surface. The results depicted in Figure 5 suggested that there was no discernible overlapping of valence charge distributions between formaldehyde and the GH substrate in the four optimized adsorption configurations. Evidently, these findings implied that there was no substantial electronic transfer, leading to the formation of significant chemical bonds during the adsorption process. The primary adsorption mechanism was thus attributed to van der Waals interactions between molecules [51].

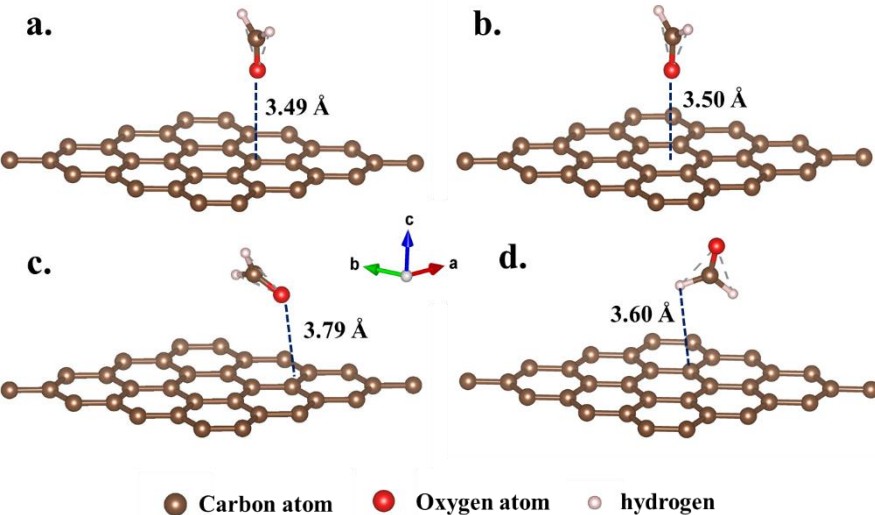

**Figure 4.** The optimized configurations of formaldehyde molecules adsorbed on GH substrate. (configurations (**a**–**d**) are calculated from the original configurations Figures S1–S4, respectively).

**Table 1.** Adsorption length and adsorption energy of formaldehyde adsorbed on GH and Fe-GH substrate.

| Adsorption Configuration | Formaldehyde Adsorbed on GH | |
| :---: | :---: | :---: |
| | Adsorption Length/Å | Adsorption Energy/eV |
| a. | 3.49 | −0.0255 |
| b. | 3.50 | −0.0230 |
| c. | 3.79 | −0.0227 |
| d. | 3.60 | −0.0282 |
| **Adsorption Configuration** | **Formaldehyde Adsorbed on Fe-GH** | |
| | Adsorption Length (O-Fe)/Å | Adsorption Energy/eV |
| a. | 1.875 | −1.280 |
| b. | 1.878 | −1.270 |
| c. | 1.882 | −1.282 |
| d. | 1.907 | −1.508 |
| e | 1.947 (H-Fe) | −0.155 |

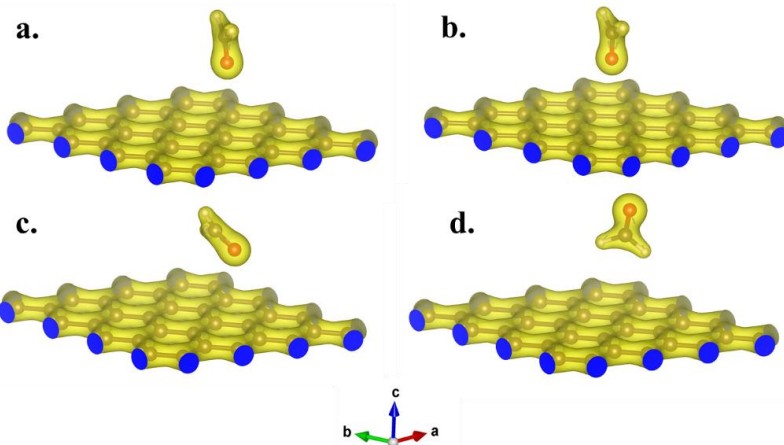

**Figure 5.** The valence charge distribution of optimized configurations of formaldehyde molecules adsorbed on GH substrate. (The distribution results of (**a**–**d**) correspond to the optimal configurations of Figure 4a–d, respectively).

### 3.3. Adsorption Properties of Formaldehyde on Fe-Doped Grapheme

Figure 6 illustrates the five optimized configurations of formaldehyde adsorption on the Fe-GH substrate. The visual inspection of Figure 6 revealed that formaldehyde molecules consistently adsorbed in proximity to doped iron atoms, with adsorption distances recorded as 1.875 Å (a. fO-Fe), 1.878 Å (b. fO-Fe), 1.882 Å (c. fO-Fe), 1.907 Å (d. fO-Fe), and 1.947 Å (e. fH-Fe), respectively. This observation unequivocally suggested that the iron atom served as the central and active site for formaldehyde adsorption on the Fe-GH substrate [52]. Additionally, in the context of the five optimized configurations (Figure 6a–e), the first four adsorption processes occurred between the oxygen atom of the formaldehyde molecule and the iron atom of the Fe-GH substrate, while the fifth process involved the hydrogen and iron atom, exhibiting no significant bonding efficiency. Hence, the active site for the adsorption of formaldehyde molecules was identified as the oxygen atom. In the fourth configuration (Figure 6d), besides oxygen atoms, carbon atoms also formed bonds with iron atoms on the Fe-GH substrate, underscoring the significance of carbon atoms in this particular adsorption process. Moreover, when compared to the GH substrate, all four adsorption distances between formaldehyde molecules and the Fe-GH substrate were below 2 Å. This observation implied that the adsorption process should be classified as chemical adsorption.

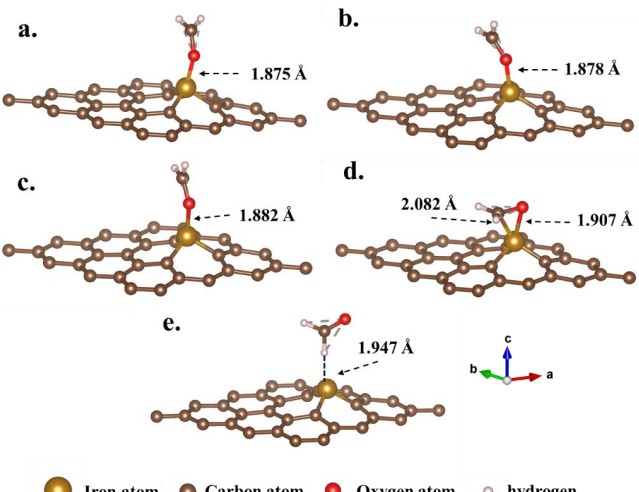

**Figure 6.** The optimized configurations of formaldehyde molecules adsorbed on Fe-GH substrate. (Configurations (**a**–**e**) are calculated from the original configurations Figures S5–S9, respectively).

Subsequently, Table 1 presents the adsorption lengths and energy calculation results for the five optimized adsorption configurations of formaldehyde on the GH substrate. Notably, the adsorption energies for the four optimized configurations were determined as −1.2799, −1.2701, −1.2824, −1.5075, and −0.1553 eV, respectively. With the exception of the fifth configuration, the negative adsorption energies signified a significant reduction in the total energy of the adsorption system following the adsorption of formaldehyde molecules on the Fe-GH substrate. Such a reduction could be unequivocally attributed to the formation of stable chemical bonds between formaldehyde molecules and the Fe-GH substrate during the adsorption process [53,54]. The notably lower adsorption energy for the fifth configuration suggested that the hydrogen atoms in formaldehyde molecules could not establish a stable adsorption structure with iron atoms. Furthermore, upon comparing the adsorption energy values of the five configurations, it became evident that the fourth adsorption configuration (Figure 6d) exhibited the highest adsorption energy, indicative of the lowest energy and the most stable structure. This heightened stability could be attributed to the simultaneous involvement of oxygen and carbon atoms in the adsorption process for this particular configuration (Figure 6d). Moreover, Figure 7 depicts the valence charge distribution of the four optimized adsorption configurations. Upon examination of Figure 7, it is evident that there was observable overlap for the first four adsorption configurations, indicating partial electron transfer during the adsorption process between formaldehyde molecules and the Fe-GH substrate [55,56]. This suggests the formation of robust chemical bonds between iron and oxygen atoms, aligning with the characteristic chemical adsorption mechanism [57]. In the case of the fourth configuration (Figure 7d), carbon atoms also contributed to charge transformation. Conversely, for the fifth configuration, no overlap of charges was observed, signifying the absence of effective charge transfer. In summary, the results indicated that via the doping of iron atoms, stable chemical bonds formed during the adsorption process with formaldehyde molecules and the Fe-GH substrate. This resulted in a reduced adsorption distance, lower adsorption energy, and effective valence electron overlap. The doped iron and oxygen atoms of the formaldehyde molecule emerged as the reaction cores in the adsorption process.

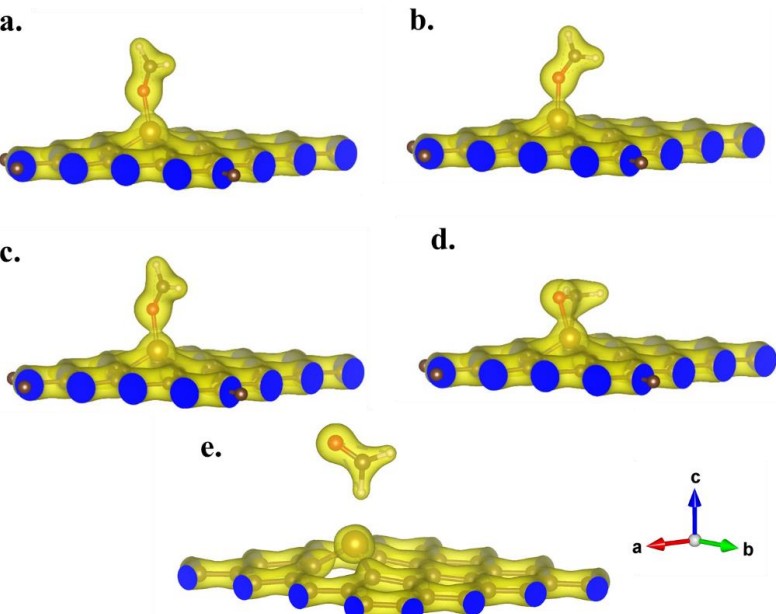

**Figure 7.** The valence charge distribution of optimized configurations of formaldehyde molecules adsorbed on Fe-GH substrate. (The distribution results of (**a–e**) correspond to the optimal configurations of Figure 6a–e, respectively).

### 3.4. Electronic Properties during the Adsorption Process

The examination of charge properties played a pivotal role in comprehending and analyzing adsorption mechanisms. In the context of the adsorption process of formaldehyde (FM) molecules on the GH substrate, the configuration depicted in Figure 3 exhibited the lowest adsorption energy, serving as the basis for subsequent research. The analysis of Bader charges proved instrumental in understanding the charge transfer dynamics during the adsorption process. Before and after the optimization of the adsorption structure, Bader charges for the formaldehyde molecule and carbon atoms near the adsorption site were calculated, with the results presented in Table 2. The data in Table 2 revealed that, prior to adsorption, the total Bader charge of the formaldehyde molecule equated to the total valence electron charge in the pseudopotential. Likewise, the Bader charge distribution for carbon atoms C1 to C4 hovered around the valence electron charge in the pseudopotential. Following the optimization of the adsorption configuration, minimal changes in Bader charge were observed for carbon atoms C1 to C4 and the formaldehyde molecule. These findings implied that there was almost negligible electron transfer during the adsorption of formaldehyde molecules on the GH substrate, aligning with the characteristics of its physical adsorption process.

**Table 2.** The Bader charge changes before and after formaldehyde molecules adsorbed on GH and Fe-GH.

| Atoms | Formaldehyde Adsorbed on GH | | |
| :---: | :---: | :---: | :---: |
|  | **Before Adsorption** | **After Adsorption** | **Change** |
| C1 | 3.962 | 3.968 | 0.00600 |
| C2 | 3.971 | 3.961 | −0.0100 |
| C3 | 3.987 | 3.981 | −0.0060 |
| C4 | 3.962 | 3.959 | −0.0030 |
| $CH_2O$ | 12 | 12.0139 | 0.0139 |
| Atoms | Formaldehyde Adsorbed on Fe-GH | | |
|  | **Before Adsorption** | **After Adsorption** | **Change** |
| C1 | 4.159 | 4.136 | −0.023 |
| C2 | 4.387 | 4.365 | −0.022 |
| C3 | 4.150 | 4.140 | −0.010 |
| Fe | 7.160 | 6.873 | −0.287 |
| $CH_2O$ | 12 | 12.338 | 0.338 |

Moreover, it was recognized that the adsorption of small gas molecules had a discernible impact on the density of states (DOS) distribution of the configuration, particularly in the vicinity of the Fermi level. The influence of chemical adsorption processes was notably more pronounced than that of physical adsorption processes. In Figure 8, a comparison of DOS calculation results is presented for the pristine GH substrate and the adsorption configuration (Figure 1). Clearly, it could be observed that the change in DOS distribution near the Fermi level for both the pristine GH substrate (Figure 8a) and the adsorption configuration (Figure 8b) was not significant. These results suggested that the adsorption process of formaldehyde molecules did not markedly affect the electronic structure and distribution of the configuration, indicating a physical adsorption mechanism [58].

Subsequently, a comprehensive analysis of charge properties concerning formaldehyde molecules absorbed on Fe-GH was undertaken. Given that the configuration depicted in Figure 3 exhibited the lowest adsorption energy, it was selected as the basis for subsequent investigations into the electronic properties of formaldehyde molecules adsorbed on the Fe-GH substrate. Initially, Table S1 presents the Bader charge analysis results for Fe-doped graphene. In comparison with the initial charge before iron doping, a discernible decrease in Bader charge for the Fe atom and an increase for C1 to C3 carbon atoms were observed. This outcome suggested that, during the iron doping process, charge transfer occurred from the vicinity of the doping site to the graphene substrate. Nevertheless, it was noteworthy

that the distribution of valence charges near iron-doped atoms remained substantially higher than that of other carbon atoms. Subsequent to this, Table 2 details the Bader charge changes before and after the adsorption of formaldehyde molecules. In comparison with the values before adsorption, a reduction in Bader charges for the iron atom and nearby C1, and C3 carbon atoms was observed, with the most significant decrease observed in the iron atom. Simultaneously, a noticeable increase in Bader charge for the formaldehyde molecule was evident. This outcome signified that, during the adsorption process, there was a substantial charge transfer between the Fe atom as the adsorption active site and the adsorbed formaldehyde molecule [59]. The adsorbed formaldehyde molecule acquired a substantial amount of charge from the substrate, thereby enhancing the stability of the adsorption results and facilitating the formation of chemical bonds [60,61]. These reciprocal charge transfers ultimately culminated in the occurrence of chemical adsorption processes.

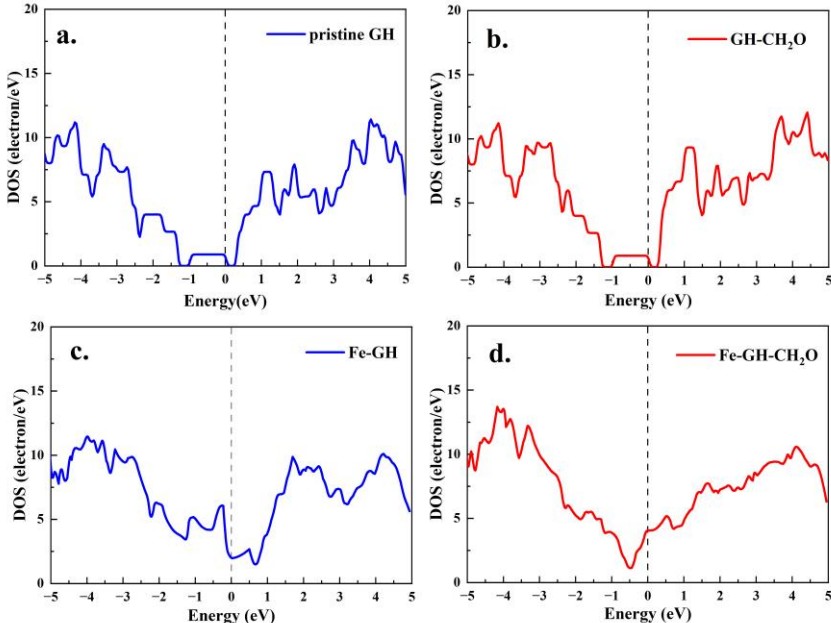

**Figure 8.** The density of states calculated result of pristine GH (**a**) and GH-CH$_2$O (**b**); the density of states calculated result of Fe-GH (**c**) and Fe-GH-CH2O (**d**).

Additionally, Figure 8c,d illustrate the electronic density of states (DOS) for the Fe-GH substrate and the formaldehyde (FH) molecular adsorption configuration. Firstly, a notable alteration in the DOS distribution around the Fermi level was observed when compared to Figure 8a, indicating that the doping of iron atoms had a considerable impact on the electronic structure near the Fermi level [62]. Furthermore, in contrast to Figure 8a–d demonstrate substantial changes in the density of state distribution around the Fermi level. This implied that, unlike the physical adsorption process depicted in Figure 8a,b, the chemical adsorption process of formaldehyde molecules on the Fe-GH substrate induced significant modifications in the charge distribution structure of the entire adsorption configuration via charge transfer. Figure 9a,b present the calculated results of the partial density of states (p-DOS) for Fe-s, Fe-p, and Fe-d orbitals before and after formaldehyde (FH) molecule adsorption. It was apparent that Fe-p and Fe-d orbitals exhibited a robust distribution near the Fermi level before adsorption. Subsequently, after FH molecule adsorption (Figure 9b), there was a pronounced alteration in the p-DOS of Fe-s, Fe-p, and Fe-d orbitals near the Fermi level. This observation aligned with the adsorption active site of iron atoms, suggesting that the electronic states of Fe-p and Fe-d orbitals played a pivotal role in facilitating the formation of adsorption bonds between the Fe-GH substrate and formaldehyde molecules [63].

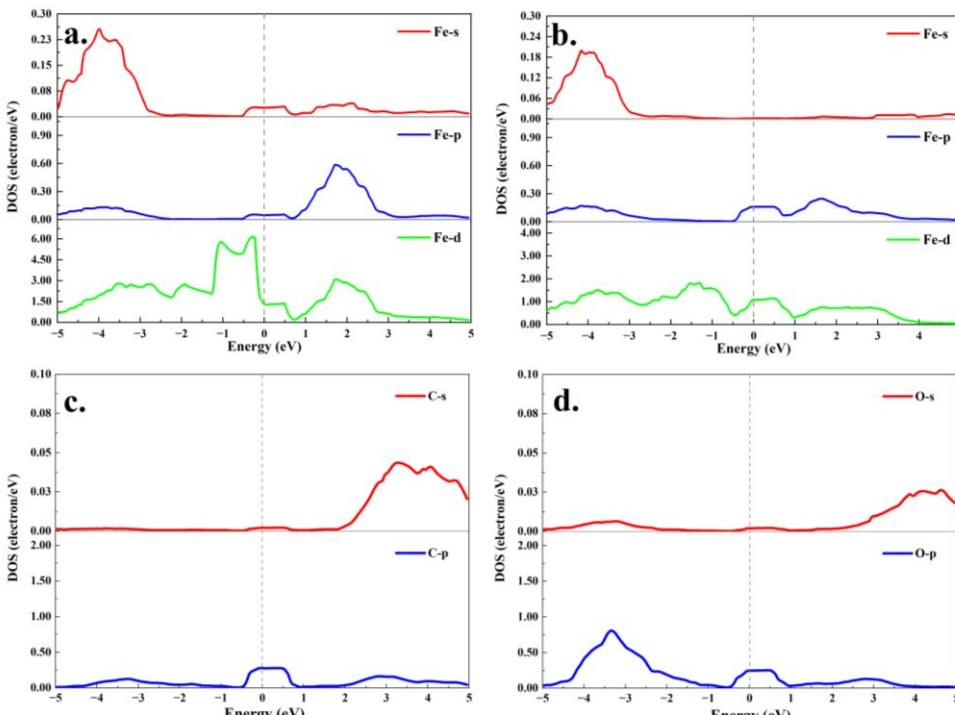

**Figure 9.** The iron partial density of states calculated result of Fe-GH (**a**) and Fe-GH-CH$_2$O (**b**); the carbon (**c**) and oxygen (**d**) partial density of states calculated result of formaldehyde molecules after adsorption.

Indeed, aside from the adsorbing substrate, the charge distribution of adsorbed gas molecules also exerted an influence on the density of states of the entire configuration. Illustrated in Figure 9c,d is the partial density of states (p-DOS) calculations for carbon (C-s and C-p) and oxygen (O-s and O-p) after the adsorption of formaldehyde molecules. Figure 9c,d reveal that the electrons in the s orbitals of carbon and oxygen atoms were scarcely distributed near the Fermi level. In contrast, the p orbitals electrons of these two atoms exhibited a more substantial distribution in proximity to the Fermi level. This outcome suggested that the s orbital electrons of carbon and oxygen atoms in the formaldehyde molecules exerted minimal influence on the bonding process. During the chemical adsorption process, the bonding effect primarily involved electrons in the p orbital of the oxygen atom in the formaldehyde molecule. Simultaneously, the p-orbital electrons of carbon atoms underwent transfer and contributed to bonding during the adsorption process, aligning with the previously discussed results of configuration optimization (Figure 6d) and charge density analysis (Figure 7d).

Figure 10 depicts the distribution of differential valence charge density in the optimal adsorption configuration. Figure 10 reveals that the entire region of the formaldehyde molecule acquired charge during the adsorption process, while the iron atom region primarily experienced charge loss. This outcome suggested a clear charge transfer from iron atoms to formaldehyde molecules during the adsorption process, in substantial agreement with the preceding analysis results from Bader charge calculations. Simultaneously, the results from Figure 10 illustrate that the transferred charges were predominantly distributed within the region between formaldehyde molecules and iron atoms, providing further validation of the bonding effect in chemical adsorption between the two entities. The pronounced charge distribution between the oxygen atom (formaldehyde) and iron atoms, as well as between the carbon atom (formaldehyde) and iron atoms, indicated the active participation of both oxygen and carbon atoms in the formaldehyde molecules during the adsorption process. Conversely, the two hydrogen atoms in the formaldehyde molecule

did not actively engage in the adsorption process, and there was no significant change in charge intensity in their vicinity.

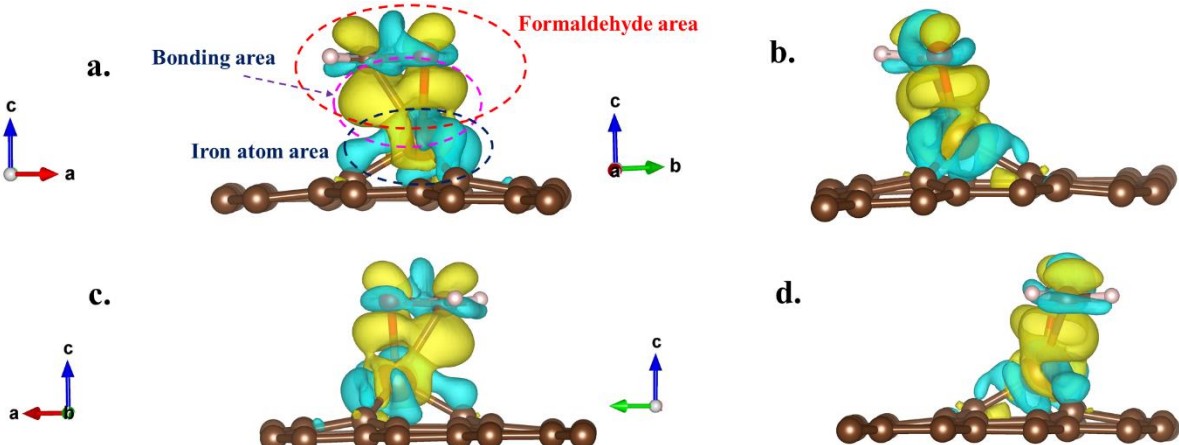

**Figure 10.** The differential valence charge density distribution of optimal adsorption configuration of formaldehyde molecules on Fe-GH substrate (Figure 7d). ((**a**–**d**) represent different observation directions, which can be determined by the lattice arrow in the bottom left corner, then yellow color represents electrons obtained; blue color represents electron lost).

In summary of the aforementioned computational outcomes, it was evident that the doping of iron atoms in graphene effectively altered the initial valence charge distribution, resulting in a spherical and concentrated distribution state near the iron atoms. This particular charge distribution proved to be instrumental in facilitating the adsorption of formaldehyde molecules on the Fe-GH surface. Notably, the oxygen atoms in formaldehyde molecules exhibited an effective exchange of charges with doped iron atoms, leading to the formation of stable chemical bonds. As carbon atoms in formaldehyde molecules are engaged in charge transfer with iron atoms, the adsorption process attained maximum adsorption energy and a highly stable chemical structure. The participation of p and d orbital electrons in iron atoms within the Fe-GH substrate, alongside p orbital electrons in oxygen and carbon atoms of formaldehyde molecules, primarily contributed to charge transfer. This interaction significantly influenced the distribution of the density of states throughout the entire adsorption configuration near the Fermi level. The electrons in these orbitals served as the primary catalyst for bonding in the adsorption process, transforming it from physical adsorption to more enduring chemical adsorption. The obvious electron transfer during the chemical adsorption process could be amplified and received by relevant electronic devices to achieve effective detection of formaldehyde molecules.

### 3.5. Optical Properties

The detection of gas molecules stands as a significant application domain for adsorption materials. The comparison of optical properties before and after gas adsorption serves as one of the simplest and most effective detection methods, exhibiting resilience to substantial alterations in material optical properties pre- and post-adsorption. Illustrated in Figure 11 are the computational outcomes of optical properties for Fe-GH and Fe-GH-$CH_2O$. Figure 11 indicates a substantial theoretical shift in optical properties before and after the adsorption of formaldehyde molecules. Notably, a significant red shift phenomenon was observed around 260 to 320 and 400 to 550 nm, while a pronounced blue shift phenomenon occurred at 550 to 680 nm. Additionally, there was a marked increase in light absorption intensity within the range of 400 to 550 nm, accompanied by a notable decrease of around 550 to 680 nm. This difference in optical properties could be amplified and transformed via optoelectronic materials, thereby achieving the detection of formaldehyde concentration. So, these discernible spectral changes offered valuable insights for detecting the adsorption of formaldehyde molecules on the Fe-GH substrate, with potential applications in the

determination of formaldehyde in gaseous environments. As we know, the current research is still in the theoretical stage of DFT calculation, and there are still many limitations in practical applications. There are potential challenges in achieving the practical application of iron-doped graphene in the field of formaldehyde detection, including material preparation, the verification of actual adsorption performance, photoelectric signal transfer, measurement accuracy, and measurement stability. However, as previously mentioned, the research results of this study are meaningful for expanding the research ideas and approaches of efficient, simple, and affordable formaldehyde detection technology.

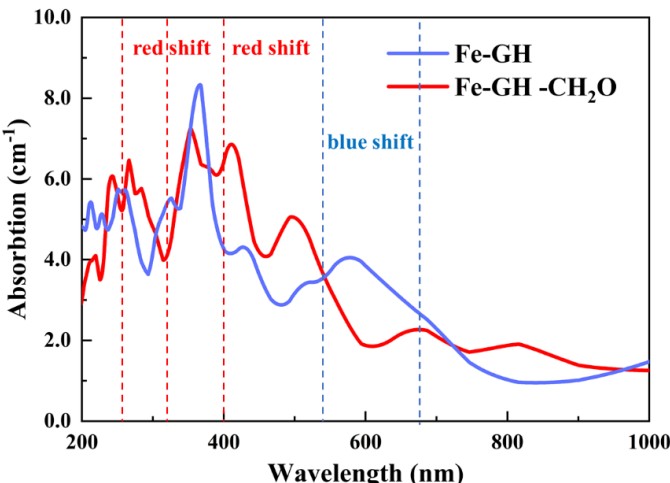

**Figure 11.** The calculation results of optical properties of Fe-GH and Fe-GH-$CH_2O$.

### 4. Conclusions

The in situ doping of graphene molecules with iron atoms demonstrated efficacy in modifying the local charge distribution proximate to the doped atoms. This modification created a conducive environment for heightened adsorption of formaldehyde molecules in the immediate vicinity. The adsorption process manifested typical attributes of chemical adsorption, characterized by augmented adsorption energy, a more stable adsorption configuration, and diminished adsorption spacing. The substantial charge transfer that ensued facilitates the establishment of chemical adsorption bonds. This involved the concurrent formation of bonds between the oxygen atom (of the formaldehyde molecule) and the doped iron atom, as well as the carbon atom (of the formaldehyde molecule) and the doped iron atom. The predominant contributors to this charge transfer were the p and d orbital electrons within the iron atoms of the Fe-GH substrate, alongside the p orbital electrons within the oxygen and carbon atoms of the formaldehyde molecules. The orchestrated transfer of charge between formaldehyde molecules and doped iron atoms assumed a pivotal role in fostering the development of chemical adsorption bonds. The discernible electron transfer during the chemical adsorption process might be amplified by macroscopic electronic devices, thereby enabling the quantification of formaldehyde concentration via electrical signals. Ultimately, substantial theoretical distinctions in optical properties existed between the configuration before and after adsorption. The noteworthy alterations in the optical properties of formaldehyde pre- and post-adsorption on the Fe-GH substrate could be translated into effective photoelectric models. These models, in turn, could serve the purpose of efficient formaldehyde detection. Consequently, the results of the density functional theory (DFT) calculations positioned iron-doped graphene as a material with promising potential for applications in the realm of formaldehyde molecular detection. Simultaneously, owing to the simplicity, cost-effectiveness, and operational ease of the adsorption method, coupled with the efficient adsorption performance and evident theoretical electronic and photoelectric effects of the Fe-GH substrate, it also exhibited significant developmental potential in the domain of large-scale formaldehyde detection.

**Supplementary Materials:** The supporting information can be downloaded at https://www.mdpi.com/article/10.3390/coatings13122034/s1. Figure S1: The original configuration (a) of formaldehyde molecules adsorbed on GH substrate; Figure S2: The original configuration (b) of formaldehyde molecules adsorbed on GH substrate; Figure S3: The original configuration (c) of formaldehyde molecules adsorbed on GH substrate; Figure S4: The original configuration (d) of formaldehyde molecules adsorbed on GH substrate; Figure S5: The original configuration (a) of formaldehyde molecules adsorbed on Fe-GH substrate; Figure S6: The original configuration (b) of formaldehyde molecules adsorbed on Fe-GH substrate; Figure S7: The original configuration (c) of formaldehyde molecules adsorbed on Fe-GH substrate; Figure S8: The original configuration (d) of formaldehyde molecules adsorbed on Fe-GH substrate; Figure S9: The original configuration (e) of formaldehyde molecules adsorbed on Fe-GH substrate; Table S1: Bader charge before and after iron doping on GH substrate.

**Author Contributions:** Conceptualization, X.Z. and C.C.; methodology, T.C.; software, Y.Y. and J.L.; data curation, J.Z., B.H., X.X., and M.W.; writing—original draft preparation, X.Z.; writing—review and editing, C.C.; visualization, T.C.; supervision, X.Z.; project administration, C.C.; funding acquisition, C.C. All authors have read and agreed to the published version of the manuscript.

**Funding:** This work is supported by Zhenjiang City 2021 key research and development project (Social Development) under Grant (SH2021020).

**Institutional Review Board Statement:** Not applicable.

**Informed Consent Statement:** Not applicable.

**Data Availability Statement:** Data are contained within the article.

**Conflicts of Interest:** The authors declare that they have no known competing financial interests or personal relationships that could have appeared to influence the work reported in this paper.

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
