# Peer review of "Optimizing the Local Charge of Graphene via Iron Doping to Promote the Adsorption of Formaldehyde Molecules—A Density Functional Theory Study"

_coatings, doi:10.3390/coatings13122034_

Round 1
Reviewer 1 Report
Comments and Suggestions for Authors
Zhang et al. provide DFT study for optimizing the local charge of graphene by iron doping to promote the adsorption of formaldehyde molecules. For study it was used PBE with GGA as it is implemented in VASP package. The authors provide the formula for computing formation energy of iron-doped graphene, adsorption energy of formaldehyde molecules on the original and Fe doped graphene. The study is based on calculations itself as it is. Thus, the manuscript must be improved before publication:
Line 40-42. It is unclear how quality standard is related to the study as the manuscript didn’t provide statistical data between different countries. Also, the sentence must cite scientific literature as it is providing other data.
The first paragraph is dedicated to Farmaldehyde and the second one is dedicated to graphene. Thus, it is difficult understand the research idea why these two molecules must be analyzed together. The Introduction must be rewritten in order at fist to clarify the importance of the research.
The figure study object appears on page 2 (132 line) and it references to supplementary material. The Introduction must introduce the study object by figures also.
Computational methods sections line 132-150 is unclear. Is it results? It must be moved and updated.
157,164, 211,212, 181 lines in results havn’t atom numbers while in line 274 the atom numbers is referenced. The manuscript must checked and all atoms must be labeled.
The manuscript has 11 figures and they repeat each other. It is difficult to follow. I would expect Figures 6, 7, 8, 9 can combined into 1 or 2 figures.
The manuscript has 5 tables which repeat each other according their structure. Moreover, Table 2 is different. The data provided is raw data and they must be revised, moved to supplementary data. The tables clearly can be revised as 1-2 tables.
I would like to see data importance for experimental data from literature and discussions about it.
Author Response
Reviewer 1
(1) Line 40-42. It is unclear how quality standard is related to the study as the manuscript didn’t provide statistical data between different countries. Also, the sentence must cite scientific literature as it is providing other data.
Reply: Thank you very much for your comments. The World Health Organization has designated formaldehyde as a class 1 carcinogen and categorized it as a toxic and harmful water pollutant (class 1), and the relevant literatures have been cited in our manuscript.
(2) The first paragraph is dedicated to Farmaldehyde and the second one is dedicated to graphene. Thus, it is difficult understand the research idea why these two molecules must be analyzed together. The Introduction must be rewritten in order at fist to clarify the importance of the research.
Reply: Thank you very much for your comments. In the revised manuscript, the introduction section has been optimized and strengthened, and the modified section has been highlighted
(3) The figure study object appears on page 2 (132 line) and it references to supplementary material. The Introduction must introduce the study object by figures also.
Reply: Thank you very much for your comments. According to the comments of reviewer, the figure in supplementary material has been added in the main manuscript.
(4) Computational methods sections line 132-150 is unclear. Is it results? It must be moved and updated.
Reply: Thank you very much for your comments. Computational methods have been further explained in this section. The mentioned part (line 132-150) introduced the original configuration before calculation.
(5) 157,164, 211,212, 181 lines in results haven’t atom numbers while in line 274 the atom numbers is referenced. The manuscript must checked and all atoms must be labeled.
Reply: Thank you very much for your comments. We have redefined the atomic numbers near the occurrence of adsorption in the captions of new Figure1 and 2. The mentioned parts (157,164, 211,212, 181 lines) have also been optimized.
(6) The manuscript has 11 figures and they repeat each other. It is difficult to follow. I would expect Figures 6, 7, 8, 9 can combined into 1 or 2 figures.
Reply: According to the comments of reviewer, original Figures 6, 7, 8, 9 has been merged and optimized.
(7) The manuscript has 5 tables which repeat each other according their structure. Moreover, Table 2 is different. The data provided is raw data and they must be revised, moved to supplementary data. The tables clearly can be revised as 1-2 tables.
Reply: Thank you very much for your comments. The data in the related tables has been revised (retain three significant digits after the decimal point), and original Tables has been merged and optimized.
(8) I would like to see data importance for experimental data from literature and discussions about it.
Reply: Thank you very much for your comments. This research is mainly about theoretical calculations, and the results offer valuable insights for detecting the adsorption of formaldehyde molecules on the Fe-GH substrate, with potential applications in the determination of formaldehyde in gaseous environments. As we know, the current research is still in the theoretical stage of DFT calculation, and there are still many limitations in practical experiments and application, including material preparation, verification of actual adsorption performance, photoelectric signal transfer, measurement accuracy, and measurement stability. Therefore, it would be currently difficult for us to discuss the theoretical calculation results with actual experiment data.
Reviewer 2 Report
Comments and Suggestions for Authors
In this work, Zhang et al. report a computational (DFT) study on the adsorption of formaldehyde on Fe-doped graphene followed by an analysis of the change in optical and electronical properties of the material after such adsorption. In brief, I did not find this work suitable for publication in Coatings for the following reasons:
1) The study has nothing to do with the journal "Coatings". Whereas in some cases graphene can be used as a coating, this study is by far not related with such an application. This study is just about the theoretical interaction of iron-doped graphene with formaldehyde and does not specifically refer to any kind of coating. This work should have been submitted to different journals such as Materials or Molecules (but see comment 2)
2) The theoretical investigation of the interaction between Fe-doped graphene and formaldehyde has been reported in the past (Molecular Catalysis, 2022, 528, 112516) in a more comprehensive study involving also other metals and a more systematic determination of the free energy of adsorption at different temperatures. The authors of the current manuscript simply obtained the same result as in the previous publication, i.e., that the adsorption of formaldehyde on graphene is thermodynamically favorable (Gibbs free energy change <0).
Author Response
Reviewer 2
(1) The study has nothing to do with the journal "Coatings". Whereas in some cases graphene can be used as a coating, this study is by far not related with such an application. This study is just about the theoretical interaction of iron-doped graphene with formaldehyde and does not specifically refer to any kind of coating. This work should have been submitted to different journals such as Materials or Molecules (but see comment 2)
Reply: Thank you very much for your comments. As the scope of “Coating” includes “Theoretical and computational modeling of surface and interfaces”, it could be believed that this manuscript about theoretical DFT calculations should be suitable for publication in “Coating”.
(2) The theoretical investigation of the interaction between Fe-doped graphene and formaldehyde has been reported in the past (Molecular Catalysis, 2022, 528, 112516) in a more comprehensive study involving also other metals and a more systematic determination of the free energy of adsorption at different temperatures. The authors of the current manuscript simply obtained the same result as in the previous publication, i.e., that the adsorption of formaldehyde on graphene is thermodynamically favorable (Gibbs free energy change <0).
Reply: Thank you very much for your comments. The theoretical investigation of catalytic interaction between Fe-doped graphene and formaldehyde has indeed been reported in the past (Molecular Catalysis, 2022, 528, 112516), and we cited this literature in our manuscript. As we know, the research in above article focused more on the process of catalytic oxidation for formaldehyde molecules. Relatively speaking, our research focuses more on the optimal configuration of formaldehyde adsorption, the charge transfer process in the optimal adsorption process, and the induced changes in optical properties. We can believe that the study of formaldehyde adsorption on Fe doped grapheme in our manuscript should be a more comprehensive supplement and in-depth study on only adsorption process.
Reviewer 3 Report
Comments and Suggestions for Authors
The article titled "Optimizing the Local Charge of Graphene by Iron Doping to Promote the Adsorption of Formaldehyde Molecules - Density Functional Theory Study" delves into the utilization of theoretical first-principles density functional technology to enhance the efficiency of Fe-doped graphene in formaldehyde adsorption. The study reveals that Fe-doped graphene serves as a stable and effective doping structure, resulting in a significant shift in valence charge distribution around the doped iron atom. This altered charge distribution facilitates the chemical adsorption process, leading to reduced adsorption spacing and increased adsorption energy. Notably, there is evident charge transfer between carbon (formaldehyde) and iron atoms, as well as between oxygen (formaldehyde) and iron atoms throughout the chemical adsorption process. The formation of adsorption bonds involves the p-orbital electrons of carbon and oxygen atoms, along with the p- and d-orbital electrons of iron atoms. The study concludes that the Fe-doped graphene material exhibits promising applications in formaldehyde molecular detection, marked by significant theoretical disparities in optical properties before and after the adsorption process. Overall, the article shows promise for publication in the Journal of Coatings, I recommend major revision of the manuscript after addressing the following comments:
1. The abstract is comprehensive but could be strengthened by providing a brief mention of the key computational parameters or methodology used in the density functional theory study.
- English and typos mistakes were noticed in the paper, please check the whole paper.
2. In the introduction, please highlight the significance of formaldehyde detection and the need for efficient adsorption materials.
3. The paper would benefit from a more comprehensive discussion on the limitations and potential challenges of using Fe-doped graphene for formaldehyde adsorption.
4. The authors should provide more information on the experimental validation of their theoretical findings, such as adsorption experiments or spectroscopic characterization.
5. Compare with other adsorption materials or methods for formaldehyde detection, which would provide a better understanding of the advantages and limitations of Fe-doped graphene.
6. authors are suggested to discuss the potential environmental and health implications of using Fe-doped graphene for formaldehyde adsorption, considering the toxicity of both formaldehyde and iron, if possible.
7. Authors are suggested to refer to recent literature in graphene.
8. Authors are recommended to provide more information on the accuracy and reliability of the DFT calculations, such as benchmarking against experimental data or other theoretical methods.
9. Conclusion require further elaborations about the main findings and their implications for formaldehyde detection.
10. The authors are suggested to discuss the potential scalability and practicality of using Fe-doped graphene for large-scale formaldehyde detection applications.
Comments on the Quality of English Language
Moderate english corrections are required.
Author Response
Reviewer 3
(1) The abstract is comprehensive but could be strengthened by providing a brief mention of the key computational parameters or methodology used in the density functional theory study.
Reply: Thank you very much for your comments. We have strengthened the abstract by providing a brief mention of the key computational parameters and methodology used in the density functional theory study.
- English and typos mistakes were noticed in the paper, please check the whole paper.
Reply: Thank you very much for your comments. English and typos mistakes have checked in the whole manuscript.
(2) In the introduction, please highlight the significance of formaldehyde detection and the need for efficient adsorption materials.
Reply: Thank you very much for your comments. The significance of formaldehyde detection and the need for efficient adsorption materials have been emphasized in the introduction section.
(3) The paper would benefit from a more comprehensive discussion on the limitations and potential challenges of using Fe-doped graphene for formaldehyde adsorption.
Reply: Thank you very much for your comments. This research is mainly about theoretical calculations, and the results offer valuable insights for detecting the adsorption of formaldehyde molecules on the Fe-GH substrate, with potential applications in the determination of formaldehyde in gaseous environments. As we know, the current research is still in the theoretical stage of DFT calculation, and there are still many limitations in practical applications. There are potential challenges in achieving the practical application of iron doped graphene in the field of formaldehyde detection, including material preparation, verification of actual adsorption performance, photoelectric signal transfer, measurement accuracy, and measurement stability. However, as we said, the research results of this study are meaningful for expanding the research ideas and approaches of efficient, simple, and affordable formaldehyde detection technology. The detailed explanation is presented in section 3.4 “Optical properties”.
(4) The authors should provide more information on the experimental validation of their theoretical findings, such as adsorption experiments or spectroscopic characterization.
Reply: Thank you very much for your comments. This is a very good suggestion. But, our manuscript is mainly about theoretical calculations, and focuses more on the optimal configuration of formaldehyde adsorption, the charge transfer process in the optimal adsorption process and the induced changes in optical properties. In future research, we will consider conducting actual adsorption experiments and spectroscopic characterization.
(5) Compare with other adsorption materials or methods for formaldehyde detection, which would provide a better understanding of the advantages and limitations of Fe-doped graphene.
Reply: Thank you very much for your comments. As we know, the detection and mitigation of formaldehyde emerge as critical research areas in the realm of environmental protection. Currently, mature formaldehyde detection methods include gas chromatography, polarography and fluorometry method. Due to the large equipment size, complex detection methods, and high cost, all these methods were not suitable for long-term, large-scale, and simple indoor formaldehyde detection. Therefore, the development of formaldehyde detection technology with simple equipment, convenient operation, and low cost is worth studying. The detailed contents are presented in section “Introduction”.
(6) Authors are suggested to discuss the potential environmental and health implications of using Fe-doped graphene for formaldehyde adsorption, considering the toxicity of both formaldehyde and iron, if possible.
Reply: Thank you very much for your comments. This is a good suggestion. Using Fe-doped graphene for formaldehyde adsorption does have potential environmental and health impacts. Although graphene and metal doped graphene are low toxicity or non-toxic materials, how to fix it on the gas detector and ensure it does not leak requires more research. In addition, more research is needed on the treatment of adsorbents after adsorption, and how to resolve them to prevent their accumulation on gas detectors, which can have an impact on the environment and human health. But, our manuscript is mainly about theoretical calculations, and focuses more on the optimal configuration of formaldehyde adsorption, the charge transfer in the optimal adsorption process. The toxicity of the environment cannot be obtained through theoretical calculations, and that requires further biological experiments to obtain the related results. We will strengthen our research in this area in the future.
(7) Authors are suggested to refer to recent literature in graphene.
Reply: Thank you for your comments. Recent literatures in grapheme have been added in the manuscript.
(8) Authors are recommended to provide more information on the accuracy and reliability of the DFT calculations, such as benchmarking against experimental data or other theoretical methods.
Reply: Thank you for your comments. More information on the accuracy and reliability of the DFT calculations has been provided in section “Computational Methods and models”.
(9) Conclusion requires further elaborations about the main findings and their implications for formaldehyde detection.
Reply: Thank you for your comments. Conclusion has been rewritten and the content about main findings and their implications for formaldehyde detection have been added.
(10) The authors are suggested to discuss the potential scalability and practicality of using Fe-doped graphene for large-scale formaldehyde detection applications.
Reply: Thank you for your comments. The potential scalability and practicality of using Fe-doped graphene for large-scale formaldehyde detection applications has been added in section 4 “Conclusions”.
Round 2
Reviewer 1 Report
Comments and Suggestions for Authors
The authors improved the paper and it is much clearer the study. I think it can be published as it is.
Reviewer 2 Report
Comments and Suggestions for Authors
Considering the response of the authors, and although the novelty of this work is seriously affected by the previous publication mentioned in my previous review, I found the revised version of the manuscript strongly improved upon recommendation of the reviewers and sufficiently different from the previous work. Therefore I believe that this work is acceptable in Coatings
Reviewer 3 Report
Comments and Suggestions for Authors
Authors have responded to all comments. The manuscript revised and got improved and hence the current version of the manuscript can be accepted in Coatings.
Comments on the Quality of English LanguageEnglish of the manuscript got improved, English proof might improve the presentation during the proofreading stage after acceptance.